# Incubator-based Sound Attenuation: Active Noise Control In A Simulated Clinical Environment

**George Hutchinson**[1]*, **Lilin Du**[1], **Kaashif Ahmad**[2,3]

**1** Invictus Medical, Inc., San Antonio, Texas, United States of America, **2** Pediatrix Medical Group, San Antonio, Texas, United States of America, **3** Baylor College of Medicine, San Antonio, Texas, United States of America

* ghutchinson@invictusmed.com

## Abstract

### Objective

Noise in the neonatal intensive care unit can be detrimental to the health of the hospitalized infant. Means of reducing that noise include staff training, warning lights, and ear coverings, all of which have had limited success. Single family rooms, while an improvement, also expose the hospitalized infant to the same device alarms and mechanical noises found in open bay units.

### Methods

We evaluated a non-contact incubator-based active noise control device (Neoasis™, Invictus Medical, San Antonio, Texas) in a simulated neonatal intensive care unit (NICU) setting to determine whether it could effectively reduce the noise exposure of infants within an incubator. In the NICU simulation center, we generated a series of clinically appropriate sound sequences with bedside medical devices such as a patient monitor and fluid infusion devices, hospital air handling systems, and device mechanical sounds. A microphone-equipped infant mannequin was oriented within an incubator. Measurements were made with the microphones with the Neoasis™ deactivated and activated.

### Results

The active noise control device decreased sound pressure levels for certain alarm sounds by as much as 14.4 dB (a 5.2-fold reduction in sound pressure) at the alarm tone's primary frequency. Frequencies below the 2 kHz octave band were more effectively attenuated than frequencies at or above the 2 kHz octave band. Background noise levels below 40 dBA were essentially not impacted by the active noise control device.

**Data Availability Statement:** All data files are available from https://doi.org/10.7910/DVN/SE3BJW.

**Funding:** This work was supported in part by the National Institutes of Health, grant 1R43DC018464

(GH). Invictus Medical (invictusmed.com) provided the equipment used in this study (GH, LD). Invictus Medical provided support in the form of salaries for authors GH and LD. GH is the Chief Executive Officer and the Chief Scientific Officer of this company. Invictus Medical did not have any role in the study design, data collection and analysis, decision to publish, or preparation of the manuscript. The specific roles of these authors are articulated in the 'author contributions' section. Beyond what's noted in the Author Contributions section, neither the board of directors nor the investors of the funder has had any influence on any aspect of this research.

**Competing interests:** The authors have read the journal's policy and have the following competing interests: GH is a paid employee, board member, and shareholder of Invictus Medical, and is a named inventor of patent applications assigned to the company manufacturing the equipment used in this study. LD is an employee and shareholder of the company manufacturing the equipment used in this study, and is also a named inventor of patent applications assigned to the company manufacturing the equipment used in this study. The authors would like to declare the following patents/patent applications associated with this research: US10410619, Active Noise Control Microphone Array. This does not alter our adherence to PLOS ONE policies on sharing data and materials.

## Conclusions

The active noise control device further reduces noise inside infant incubators. Device safety and potential health benefits of the quieter environment should be verified in a clinical setting.

## Introduction

For preterm infants, the mission of neonatal intensive care unit (NICU) care is to support healthy infant development in the extrauterine environment with minimal mortality. Technological advances in neonatal intensive care have improved infant survival over time [1–8]. The NICU clinical team must provide support of basic functions including temperature and humidity control, nutritional support, and more. A critical component of healthy infant development is limiting the noxious noise to which the patient is exposed [9–13] while providing appropriate aural stimulation to promote brain and language development [14,15].

Full-term newborn infants are sufficiently developed to cope with environmental stressors such as noise and light [11,16]. However, the central nervous system of a preterm infant is ill-prepared to cope with the extrauterine environment. While stressors such as light can be attenuated by practices such as dimming overhead lights or covering an incubator with a blanket, noise is not as easily addressed.

Noise levels in NICUs have been shown to be consistently louder than guidelines provided by the American Academy of Pediatrics (AAP) [17–21]. These guidelines stipulate that the noise levels that the hospitalized infants are exposed to should not exceed 45 dBA, averaged over one hour and should not exceed a maximal level of 65 dBA averaged over one second [22]. Noise measured both inside and outside an incubator show guidelines are frequently exceeded throughout the day [18].

We sought to develop a solution to the problem of excess noise exposure for the premature infant in an incubator that would not require direct patient contact, adhesives on skin, or ongoing nursing to alleviate. The device developed was evaluated in a realistic NICU environment.

## Materials and methods

### Study design & setting

Experiments were conducted in the NICU simulation laboratory at The Children's Hospital of San Antonio (San Antonio, Texas) using bedside critical care equipment. The neonatal critical care equipment was arrayed around an infant incubator (Giraffe OmniBed, GE Healthcare, Waukesha, Wisconsin) in the typical manner. Alarm volumes were set to clinically appropriate levels.

The critical care equipment included a patient monitor (IntelliVue® MX450, Philips Healthcare, Andover, Massachusetts), a syringe pump (Medfusion™ 2001, Smiths Medical, Minneapolis, Minnesota), infusion pump (BD Alaris™ Pump Module, Becton, Dickinson and Company, Franklin Lakes, New Jersey), a bubble continuous positive airway pressure (CPAP) ventilatory support device (Fisher Paykel, Auckland, New Zealand), and a ventilator (Maquet Servo-I, Getinge Group, Gothenburg, Sweden). These devices and the hospital air handling system were used to generate 10 clinically realistic sound sequences for testing (Table 1) and each sequence was repeated for five trials. The order of the condition (i.e., attenuation device

**Table 1. NICU sound sequences used for testing.**

| Sequence # | Primary Noise | | | Background Noise | | Combined |
|---|---|---|---|---|---|---|
| | Noise Source | % time active | Tones, (decreasing order of amplitude) (Hz) | Noise Sources | % time active | $L_{eq}$ 1 min (dBA) |
| 1 | Patient monitor alarm (high priority) | 50 | 960<br>2,880 | Bubble CPAP, Hospital air handling system | 100 | 69 |
| 2 | Patient monitor alarm (medium priority) | 100 | 1,440<br>480<br>960<br>1,920 | Bubble CPAP, Hospital air handling system | 100 | 72 |
| 3 | Syringe pump end volume alarm (low priority) | 8.3 | 2,761 | Bubble CPAP, Hospital air handling system | 100 | 57 |
| 4 | Syringe pump occlusion alarm (high priority) | 8.3 | 2,745 | Bubble CPAP, Hospital air handling system | 100 | 61 |
| 5 | Patient monitor high alarm & Syringe pump occlusion alarm (high priority) | 50 & 8.3 | 960<br>2,880 | Bubble CPAP, Hospital air handling system | 100 | 67 |
| 6 | Ventilator high pressure alarm (high priority) | 8.3 | 384<br>1,173<br>783 | Ventilator mechanical noise, Hospital air handling system | 100 | 56 |
| 7 | Ventilator respiration rate alarm (medium priority) | 8.3 | 384<br>1,173<br>783 | Ventilator mechanical noise, Hospital air handling system | 100 | 52 |
| 8 | Infusion pump occlusion alarm (high priority) | 8.3 | 2,200<br>550<br>1,100<br>1,650 | Bubble CPAP, Hospital air handling system | 100 | 57 |
| 9 | Infusion pump end volume alarm (low priority) | 8.3 | 2,002<br>3,003<br>1,001 | Bubble CPAP, Hospital air handling system | 100 | 55 |
| 10 | Male and female voices | 33 | No prominent tones | Bubble CPAP, Hospital air handling system | 100 | 49 |

on or off) was randomized for each of the five trials and the mannequin was re-placed in the incubator between each trial.

A male and a female voice were included in two of the sound sequences. These were recorded on a digital recorder (Zoom H4n, Zoom North America, Hauppauge, New York) via an externally connected microphone (Dayton Audio EMM-6, Dayton Audio, Springboro, Ohio). Recordings were 44.1 kHz 16-bit WAV files. The WAV files were replayed through a powered studio monitor (KRK Rokit 5, Gibson Pro Audio, Chatsworth, California). Recorded voices were used to ensure a consistent signal for all trial and conditions.

Active noise control was provided by a Neoasis™ (Invictus Medical, San Antonio, Texas), active noise control device, which deployed on the infant incubator. The Neoasis™ consists of a control unit and an outside noise sensor, both positioned outside the incubator, and two speakers and a residual noise sensor positioned within the incubator (Fig 1). Active noise control utilizes the phenomenon of incident waves summing. If one incident wave is out of phase with the other, the waves cancel each other. The device measures the sound waves outside the incubator, model what the sound waves will be after passing through the incubator wall, and generates a sound wave out of phase with the modelled wave. A residual noise sensor within the incubator provides data for the system to converge on an optimum solution.

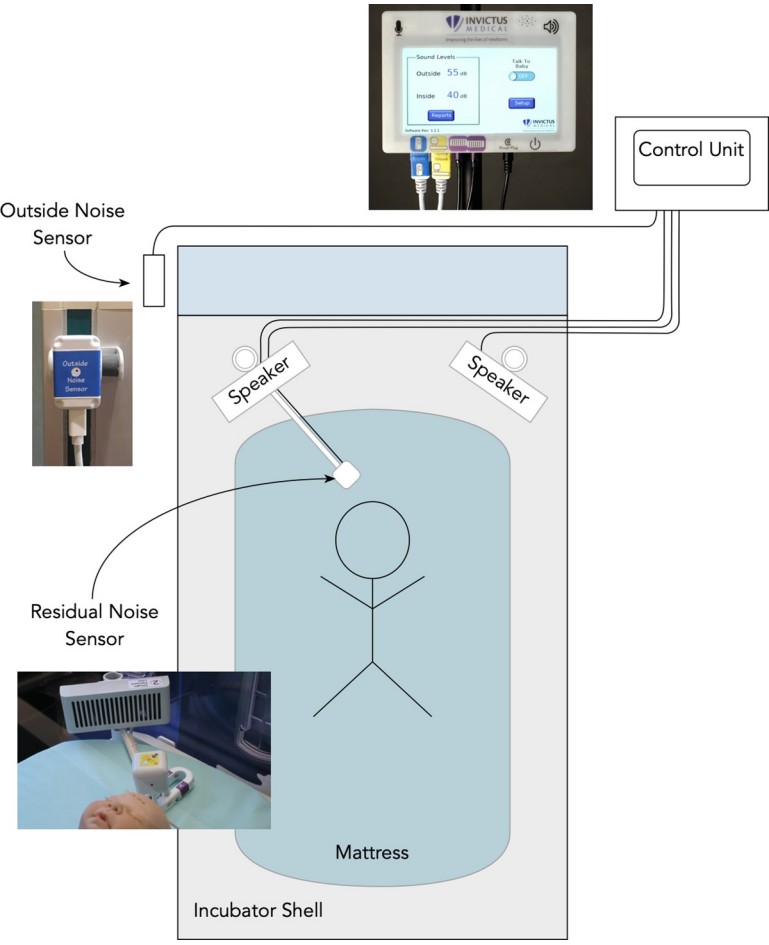

**Fig 1. Schematic of the Neoasis™ elements.** The Neoasis™ comprises a residual noise sensor, two speakers positioned inside the incubator, an outside noise sensor affixed to the outside of the incubator, and a control unit to which the other items are connected.

The outside noise sensor was affixed to the incubator on the side closest to the primary noise source, which was the alarm or voice sound in each scenario. The mannequin was placed 5 cm from the residual noise sensor per manufacturer instructions and the speakers mounted to small posts in the D-holes found in the corner of the incubator. The residual noise sensor, the two speakers, and the outside noise sensor were connected to the system's control unit. No part of the Neoasis™ device contacted the mannequin.

The mannequin was weighted and sized to resemble a 1.3 kg 29-week gestational age preterm infant [23] and was equipped with two general purpose array microphones (Model 40PP, GRAS Sound and Vibration A/S, Holte, Denmark) embedded in its head, each microphone having its sensing element positioned at an opening in the mannequin's molded ear (Fig 2). The microphones were interfaced to a computer equipped with LabVIEW™ Development System with the Sound and Vibration Toolkit (National Instruments, Austin, Texas) via a CompactDAQ Chassis containing a Sound and Vibration Input Module (National Instruments, Austin, Texas).

## Analysis

Noise attenuation was calculated according to the protocol defined by the American National Standards Institute (ANSI) standard ANSI-ASA S12.68, *Methods Of Estimating Effective*

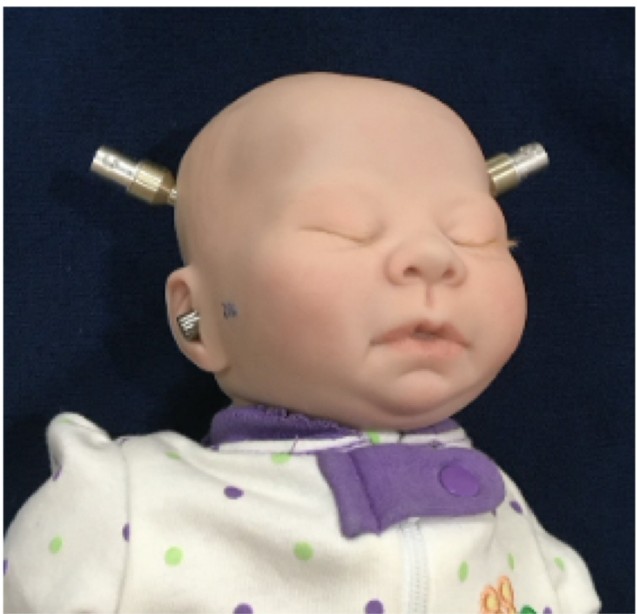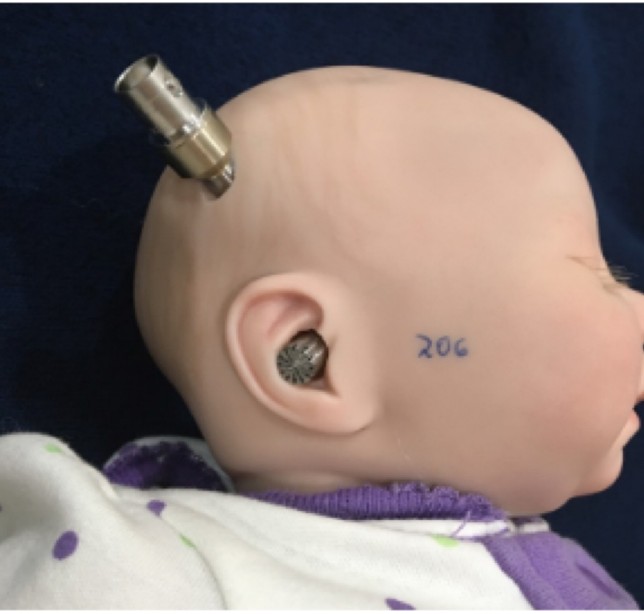

**Fig 2. Test mannequin showing microphone placement.** Holes were cut in the head of an infant mannequin, permitting the tip of each microphone to emerge from the molded auditory canal. The number written on the cheek of the mannequin indicates the calibration number of that microphone.

*A-Weighted Sound Pressure Levels When Hearing Protectors Are Worn* [24]. Two modifications had to be made to the methods described in the standard to adapt it to the test environment. First, the standard directs that users apply the noise attenuation device to themselves; since the neonate will not be self-applying the device, we modified the standard's protocol, directing that the mannequin be re-deployed in the incubator between each trial sequence. Second, the standard dictates a prescribed set of industrial noises to be used as test signals. Since industrial sounds would not be appropriate to a NICU setting, we substituted a realistic collection of sound sequences consisting of device alarms, machine noises, and hospital environmental noises (Table 1).

The ANSI standard dictates that an average pressure for each of seven octave bands (125 Hz through 8 kHz) is calculated with and without the noise attenuation device over several trials. This calculation is repeated for each ear and for each sound sequence. The difference between the noise level at an ear for an octave band represents the amount of attenuation achieved. A negative value for attenuation represents an amplification.

For sound sequences involving ventilator alarms, the ventilator was positioned by the end of the incubator away from its screen and the Neoasis™ outside noise sensor was attached to the incubator wall near the ventilator's position according to manufacturer's instructions. For all other sound sequences, the outside noise sensor was attached to the incubator's vertical rail by the unit's screen.

Per the ANSI standard, the average attenuation for all octave bands and all scenarios is calculated along with the standard deviation for the five trials and the 10 sound sequences. Since this average includes all octave bands, even those containing only low-level background noise for which little attenuation is possible or useful, the average attenuation was also calculated for all ear-sequence-octave band combinations where the unattenuated sound pressure level (SPL) was greater than 40dBA.

The Neoasis device includes a voice pass-through feature that allows a voice to be added to the cancelling soundwave to permit directed communication between the infant and the parents or caregivers. The performance of this feature was not evaluated in this study.

## Results

The amount of attenuation provided by the Neoasis™ active noise control device varied depending in part on the unattenuated SPL in a given octave band. Instances where the unattenuated SPLs were below 40 dBA, no further attenuation was achieved. For each sound sequence, SPLs for most of the octave bands for both ears were below 40 dBA and most of the sound power was focused on the frequencies of the alarm or voice tones. By way of example, the results for sound sequence 1 are shown (Fig 3). The primary tone of the sound sequence was 960 Hz, falling in the 1 kHz octave band. The attenuation of the single 960 Hz tone was 14.4 dB (a 5.2-fold decrease) while the attenuation of all sound power within the 1 kHz octave band was 11.8 dB for the left ear and 4.3 dB for the right ear. The unattenuated noise present at the left ear was greater than at the right ear, 50.7 dBA and 43.0 dBA respectively. The unattenuated sound levels of the other octave bands range from about 30 dBA to just under 40 dBA. By way of comparison, 40 dBA is comparable to a whisper at a distance of 5 ft [25].

In other noise sequences comprising different alarm tones, other phenomenon were noted. In sequences 3, 4, 9, and 10 (Table 1), the SPL of all of the octave bands were below the 40 dBA. In these scenarios, the Neoasis™ did not provide any extra attenuation to reduce the noise below a background level. In sound scenarios 2 and 8 (Table 1), the primary alarm tone was in the 2 kHz octave and the Neoasis™ provided less than 1 dB of attenuation. Of the two ears, the greater SPL was 48.7 dBA for sound scenario 2 and was 47.6 dBA for sound scenario 8.

The average sound attenuation for sound sequences, across all octave bands and both ears is 1.2 dB. The standard deviation across sound spectrums was 3.7 dB and across trials was 0.3 dB. This average includes those octave bands that only contain background noise that was not further attenuated.

Focusing on the 12 measurements where the unattenuated SPL was greater than 40 dBA (Fig 4) the amount of attenuation achieved for the tones in the 500 Hz octave band ranged from -0.2 dB to 9.8 dB, in the 1 kHz band ranged from -0.2 dB to 11.8 dB, and in the 2 kHz band ranged from -0.1 dB to 0.6 dB.

## Discussion

Our study demonstrates that an active noise control device provides attenuation within an infant incubator in a real-world simulation setting. We found substantial attenuation of as much as 14.4 dB for a commonly encountered alarm sound, equivalent to a 5.2-fold reduction in sound pressure. Sounds that were minimally transmitted through the walls of the incubator, having SPLs at and below 40 dBA, were not further attenuated. A mechanism to reduce the unnecessary noise exposure of NICU infants could have widespread application. Device alarms are consistently found to be a source of excessive noise in NICUs, [20,21,26]. Patient monitor alarms are an especially frequent contributor to the noise in NICUs, with an average of 177 alarms/patient/day [27].

Noise levels in the NICU can be quite loud [17–21] and shift changes have been observed to be the loudest periods [18]. Average noise levels in the over a rolling one hour period ranged from a low of 52 dBA to a high of 72 dBA in the NICU and from a low of 55 dBA to a high of 64 dBA within an unoccupied incubator [18]. Peak SPL measures in NICUs have been reported over 100 dBA [28,29] and within incubator as high as 88 dBA [20,21].

Attempts to improve the acoustic environment of the NICU have largely been unsuccessful, whether through education efforts [26] or sound-activated warning lights [30]. Single family rooms are being increasingly adopted in NICU design and have shown improvements in many outcomes for preterm infants [31]. The implementation of single family rooms has resulted in drops in average noise levels but the improvements have not been significant [32]

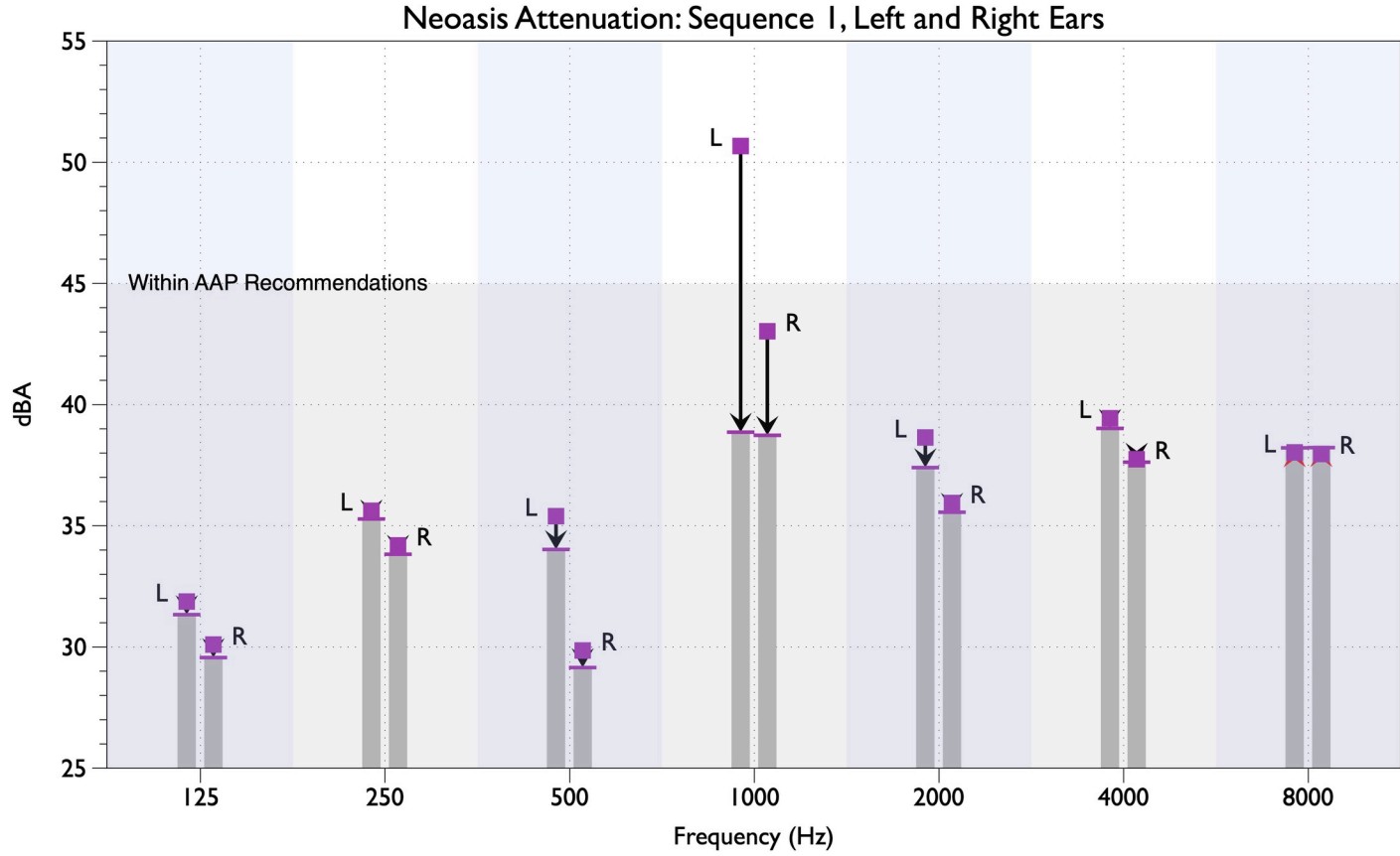

**Fig 3. Noise attenuation for noise sequence 1 for both ears and seven octave bands.** The unattenuated SPL is indicaterd by the square mark and the attenuated SPL is indicated by the bar at the top of the column. Measurements for the left and right ears are indicated by the letters "L" and "R," respectively. The shaded region on the lower portion of the graph indicates the level of the AAP guideline recommendations. The vertical shaded regions indicate each octave band.

or still remain well above the guidelines [29,32]. Given that the same monitors, ventilatory support devices and other equipment is present in the single family rooms as well as the open ward units, the continued high noise levels are not unexpected [29].

The active noise control device's having little effect on noise levels at or below 40 dBA are unlikely to have clinical relevance to the NICU patient as published guidelines focus on noise levels above 45 dBA [22,33,34]. We found that tones above 2 kHz were not addressed by the active noise control device. Active noise control systems can operate optimally over a frequency range with the maximum frequency is 30 times the minimum frequency, with decreased performance at the ends of the range [35]. With a low-end of the system set to 100 Hz, poorer performance for signals in the 2 kHz octave like those in sequences 2 and 8 is not unexpected. However, higher frequencies penetrate structures less readily than do lower frequencies [36] causing less to penetrate the incubator shell. These higher frequencies are less prevalent within the NICU than frequencies in the 500 Hz band and below [19]. We found the amplitude of the sound detected at the right ear of the mannequin was consistently lower than that found at the left ear. This likely indicates a heterogeneity of SPLs within the incubator enclosure of the incubator due to nulls of standing sound waves.

The dB scale of the measurement of SPL is logarithmic and represents the ratio between two SPLs. For example, an increase of 6 dB represents a doubling of the SPL while a 20 dB change represents a 100-fold increase. When a value is used to describe a sound or

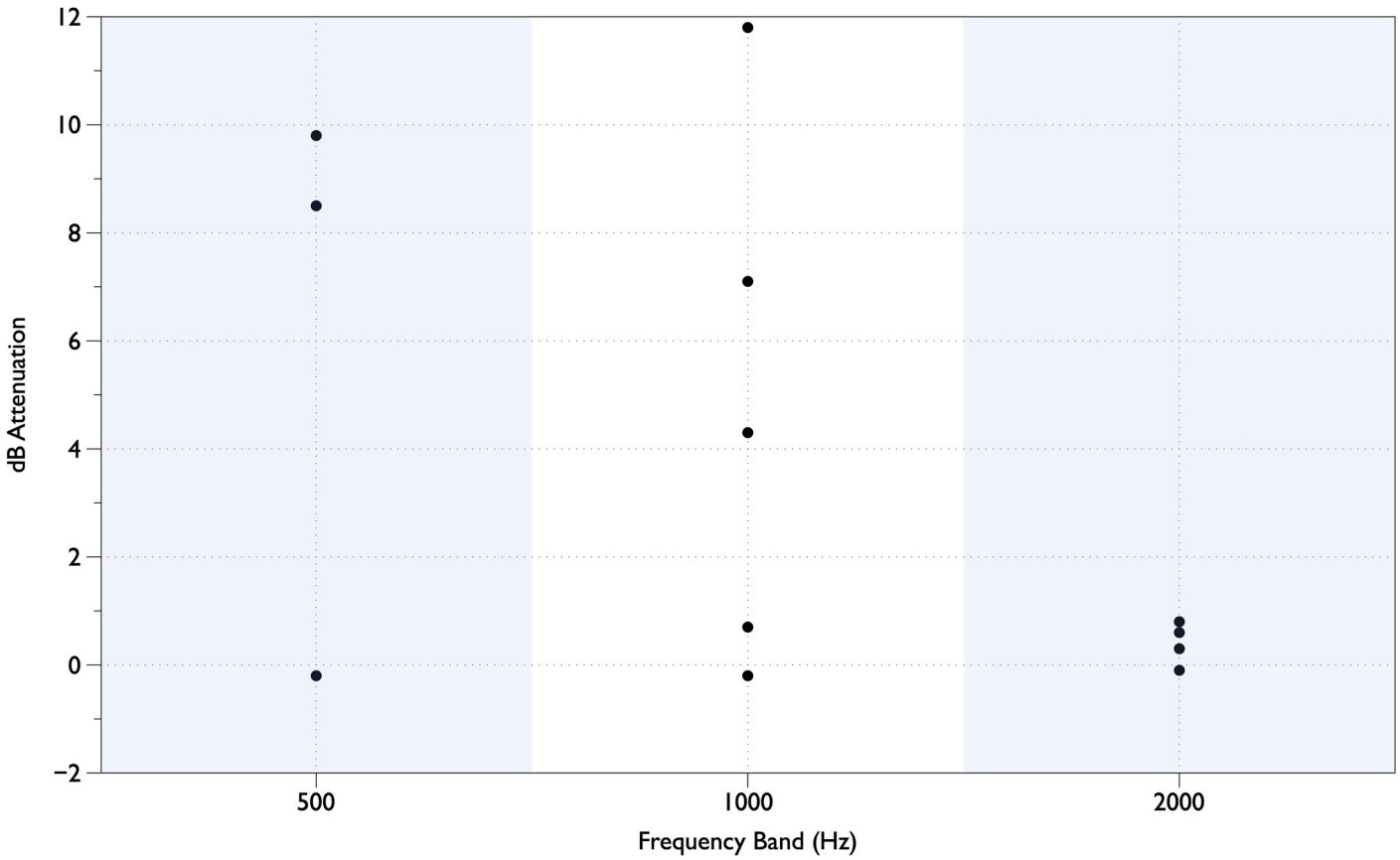

**Fig 4. Amount of attenuation for all instances when the original SPL is above 40 dBA.** The seven octave bands are shown on the x-axis and the A-weighted SPL is shown on the Y-axis. The results for the right and left ears are shown, labelled "R" and "L." The square represents the unattenuated sound and the top of the gray bar represents the attenuation achieved by the Neoasis™ system. The test sequence used comprised a high priority patient monitor alarm, a bubble CPAP device operating, and hospital environmental noise.

environment, it represents the ratio of the sound source's SPL and 0 dB, the SPL of the threshold of hearing. For example, an environmental SPL of 100 dB represents a sound whose SPL is 100,000 times more than that of the threshold of hearing.

Humans' sensitivity to sound pressure is frequency dependent, with the most sensitive range being between 1 kHz and 6 kHz and sensitivity being about 20 dB lower at 100 Hz. When dealing with situations involving human hearing, SPLs are generally weighted according to empirically-derived transfer function, referred to as A-weighting and expressed as dBA [25].

The Occupational Safety and Health Administration (OSHA) standards have been established to limit noise exposure to occupational noises to prevent worker injury [25] (see Fig 5 for examples of noises on a dBA scale). For adults, the criterion for short-term noise permissible exposure level is 90 dBA time weighted average for 15 minutes. We are not aware of studies relating time-weighted noise exposure limits for injury in infants.

Reducing noise levels for the preterm infant may have important health benefits. Studies of ear covers and ear plugs have shown that noise reduction can improve medical outcomes for preterm infants [13,37–40]. There is a concern that the adhesives required to keep ear covers on the skin of the infant are responsible for breakdown of the fragile skin of the patient [41] and that they decrease all auditory input including the directed communication needed for

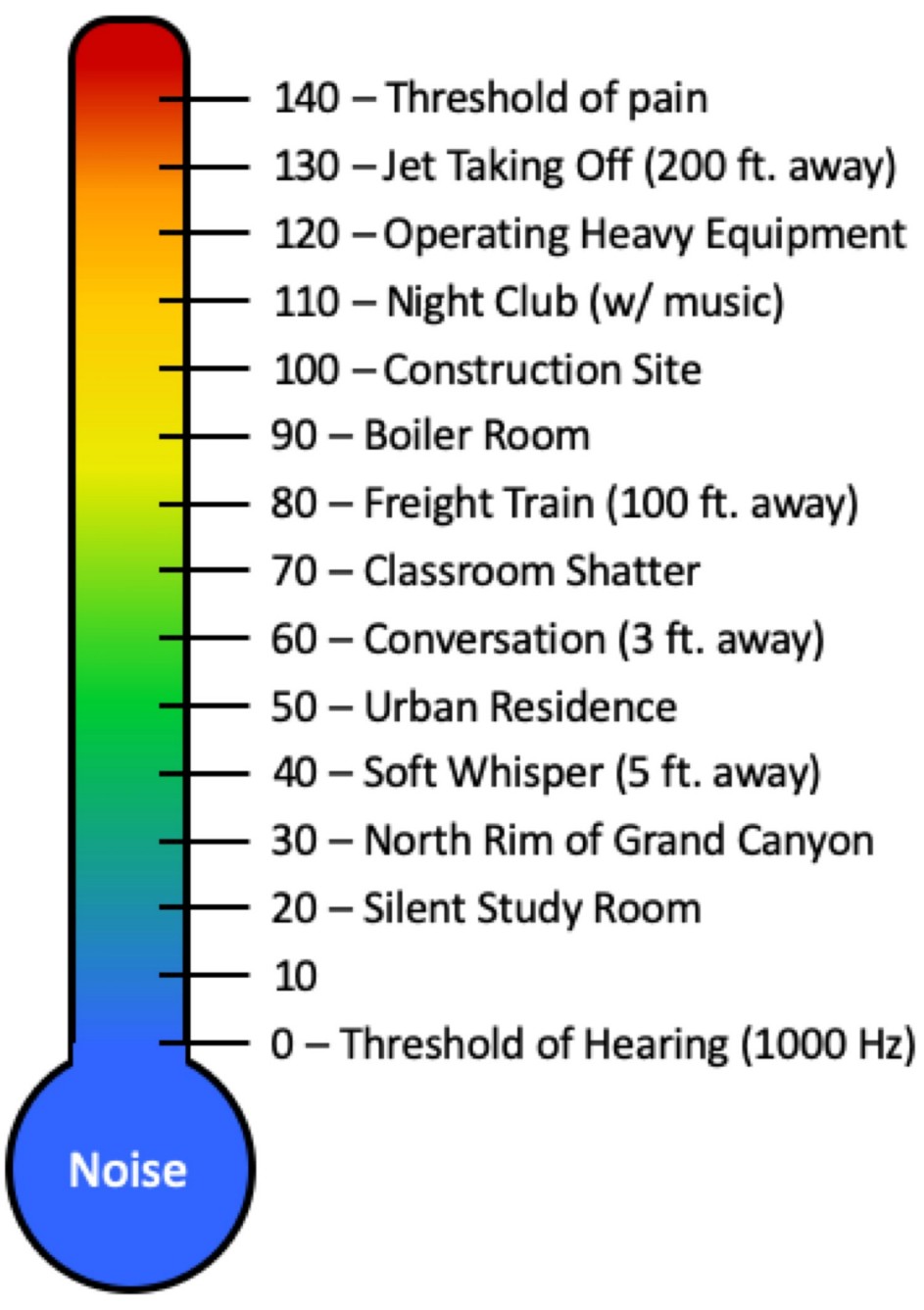

**Fig 5. Typical sound levels on the dBA scale.** Figure adapted from OSHA Technical Manual [25].

verbal development [12]. Earplugs may represent a choking hazard [13]. Improvements in weight gain due to a noise-attenuated environment have been demonstrated in randomized prospective studies of low birth weight hospitalized infants [13,39]. Adequate growth is a pre-requisite of a successful discharge from the NICU. Infants experiencing environmental stress have elevated heart and respiration rates resulting in increased oxygen consumption and caloric requirements [42].

Several limitations exist with this study. We evaluated the performance of the Neoasis™ device with a limited set of bedside devices and these devices were oriented in only one set of positions around the incubator. Only one type of incubator was used in the testing. Other bedside devices will have different alarm tones consisting of different frequency components, resulting in different levels of attenuation. It is possible that the relative position of the bedside devices, the Neoasis™ outside noise sensor, and the walls of the incubator could have an effect on the level of attenuation achieved. We also did not evaluate the performance of the voice pass-through feature of the device.

## Conclusions

Within an infant incubator in a simulated clinical environment, the Neoasis™ active noise control device was able to attenuate the device alarm sounds tested to the level of the AAP recommendations for tones in the 1 kHz octave band and below. This device may have a positive effect on noise management for the incubated infant. A clinical study is needed to verify whether the Neoasis™ will replicate the sleep and other health benefits of a reduced noise environment found in other studies.

## Acknowledgments

The authors wish to thank Drs Reese Clark and Veeral Tolia for their critical review of the manuscript. We also thank Dr Gayle Dasher and James McElroy of the simulation center at the Children's Hospital of San Antonio for their assistance in conducting the experiments in their facility and Dr Preston Wilson, Professor of Mechanical Engineering at the University of Texas at Austin for guidance in acoustics and attenuation calculations.

## Author Contributions

**Conceptualization:** George Hutchinson, Lilin Du, Kaashif Ahmad.

**Data curation:** George Hutchinson, Lilin Du.

**Formal analysis:** George Hutchinson, Lilin Du, Kaashif Ahmad.

**Funding acquisition:** George Hutchinson.

**Investigation:** George Hutchinson, Lilin Du.

**Methodology:** George Hutchinson, Lilin Du, Kaashif Ahmad.

**Project administration:** George Hutchinson.

**Resources:** George Hutchinson.

**Software:** Lilin Du.

**Supervision:** George Hutchinson, Kaashif Ahmad.

**Validation:** Lilin Du.

**Visualization:** George Hutchinson.

**Writing – original draft:** George Hutchinson.

**Writing – review & editing:** George Hutchinson, Lilin Du, Kaashif Ahmad.

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
