## [Decision Letter · Decision Letter 0]

15 May 2020

PONE-D-20-05746

Incubator-based Sound Attenuation: Active Noise Control In A Simulated Clinical Environment

PLOS ONE

Dear Dr Hutchinson,

Thank you for submitting your manuscript to PLOS ONE. After careful consideration, we feel that it has merit but does not fully meet PLOS ONE’s publication criteria as it currently stands. Therefore, we invite you to submit a revised version of the manuscript that addresses the points raised during the review process.

ACADEMIC EDITOR: 

Please make changes according to the reviewer's comments and suggestionsUnfortunately, I was only able to obtain one peer review report. I am sorry for the tardiness in evaluating the manuscript "All data files are available from " ext-link-type="uri" xlink:type="simple">invictusmed.com/nicu_sim_data" - the link does not seem to work Please make sure that the abbreviation NICU is being explained the first time it is mentioned

We would appreciate receiving your revised manuscript by Jun 29 2020 11:59PM. To enhance the reproducibility of your results, we recommend that if applicable you deposit your laboratory protocols in protocols.io, where a protocol can be assigned its own identifier (DOI) such that it can be cited independently in the future. For instructions see: http://journals.plos.org/plosone/s/submission-guidelines#loc-laboratory-protocols

We look forward to receiving your revised manuscript.

Kind regards,

Anne Lee Solevåg, M.D., Ph.D.

Academic Editor

PLOS ONE

2. Thank you for providing the following Funding Statement: 

"This work was supported in part by NIH grant 1R43DC018464 (GH). Invictus Medical (invictusmed.com) provided the equipment used in this study (GH, LD). Some authors (GH, LD) are employees of Invictus Medical and these authors played a role in the study design, data collection and analysis, decision to publish, and preparation of the manuscript. The remaining author (KA) received no specific funding for this work and played a role in the study design, data collection and analysis, decision to publish, and preparation of the manuscript."

We note that one or more of the authors is affiliated with the funding organization, indicating the funder may have had some role in the design, data collection, analysis or preparation of your manuscript for publication; in other words, the funder played an indirect role through the participation of the co-authors.

If the funding organization did not play a role in the study design, data collection and analysis, decision to publish, or preparation of the manuscript and only provided financial support in the form of authors' salaries and/or research materials, please review your statements relating to the author contributions, and ensure you have specifically and accurately indicated the role(s) that these authors had in your study in the Author Contributions section of the online submission form. Please make any necessary amendments directly within this section of the online submission form.  Please also update your Funding Statement to include the following statement: “The funder provided support in the form of salaries for authors [insert relevant initials], but did not have any additional role in the study design, data collection and analysis, decision to publish, or preparation of the manuscript. The specific roles of these authors are articulated in the ‘author contributions’ section.”

If the funding organization did have an additional role, please state and explain that role within your Funding Statement.

Please also provide an updated Competing Interests Statement declaring this commercial affiliation along with any other relevant declarations relating to employment, consultancy, patents, products in development, or marketed products, etc.  

3. Please include a copy of Table 2 which you refer to in your text on page 4.

4. We note you have included a table to which you do not refer in the text of your manuscript. Please ensure that you refer to Table 1 in your text; if accepted, production will need this reference to link the reader to the Table.

Reviewers' comments:

Reviewer's Responses to Questions

**Comments to the Author**

1. Is the manuscript technically sound, and do the data support the conclusions?

Reviewer #1: Yes

2. Has the statistical analysis been performed appropriately and rigorously? 

Reviewer #1: Yes

3. Have the authors made all data underlying the findings in their manuscript fully available?

Reviewer #1: Yes

4. Is the manuscript presented in an intelligible fashion and written in standard English?

Reviewer #1: Yes

5. Review Comments to the Author

Reviewer #1: Thank you for allowing me to review this manuscript.

The manuscript aims to examine a device which can reduce noise levels in the NICU.

While noise is an important aspect within the NICU, the effect of Noise on long-term outcomes need further studies.

In the womb, a fetus is continuously exposed to 85 dB, which is generated by the mother’s blood flow. It is worth noting that this high noise level, does not affect the development of a baby. Many guidelines recommend noise levels be 50dB, but do not consider this information.

I agree that high pitch sudden noises could have an impact of the developing brain of preterm infants, but this would require further studies.

Some data into single rooms would suggest that the reduced noise level might not support brain development either.

CPAP devices, which generate consistent white background noise, might be an advantage for the infant as it potentially masks sudden high pitch noises like alarms.

I do have a few questions/comments:

What was the Duty cycle? Could this be explained in more detail.

What statistical tests did you used?

How does the active noise canceling device work? More details would give the reader a clearer understanding about the device and how it works.

Is this device approved by health authorities like the FDA?

If you use CPAP as white background noise, would you also register the same level of dB for alarms?

6. PLOS authors have the option to publish the peer review history of their article (what does this mean?). If published, this will include your full peer review and any attached files.

Reviewer #1: Yes: Georg Schmolzer

---

## [Author Response · Author response to Decision Letter 0]

1 Jun 2020

May 23, 2020

PLoS One

Anne Lee Solevåg, M.D., Ph.D.

Academic Editor

Dear Dr. Solevåg, 

 We are happy to learn that PLOS ONE feels that our manuscript has merit and are appreciative of the opportunity to improve it in response to referee comments. We have responded to feedback point by point below: 

ACADEMIC EDITOR:

1. All data files are available from invictusmed.com/nicu_sim_data" - the link does not seem to work

- We have repaired the hyperlink and the data files are available. We apologize for the oversight.

2. Please make sure that the abbreviation NICU is being explained the first time it is mentioned

- Thank you, we have addressed this issue. 

JOURNAL REQUIREMENTS

1. Please ensure that your manuscript meets PLOS ONE's style requirements, including those for file naming

- Thank you, we have ensured the manuscript meets style requirements.

2. We note that one or more of the authors is affiliated with the funding organization, indicating the funder may have had some role in the design, data collection, analysis or preparation of your manuscript for publication; in other words, the funder played an indirect role through the participation of the co-authors.

If the funding organization did not play a role in the study design, data collection and analysis, decision to publish, or preparation of the manuscript and only provided financial support in the form of authors' salaries and/or research materials, please review your statements relating to the author contributions, and ensure you have specifically and accurately indicated the role(s) that these authors had in your study in the Author Contributions section of the online submission form. Please make any necessary amendments directly within this section of the online submission form. Please also update your Funding Statement to include the following statement: “The funder provided support in the form of salaries for authors [insert relevant initials], but did not have any additional role in the study design, data collection and analysis, decision to publish, or preparation of the manuscript. The specific roles of these authors are articulated in the ‘author contributions’ section.”

- We have included the requested statement. 

The funder provided support in the form of salaries for authors [GH, LD], but did not have any additional role in the study design, data collection and analysis, decision to publish, or preparation of the manuscript. The specific roles of these authors are articulated in the ‘author contributions’ section. 

One of the authors, Dr George Hutchinson is the Chief Executive Officer and the Chief Scientific Officer of the funder. Beyond what’s noted in the Author Contributions section, neither the board of directors nor the investors of the funder has had any influence on any aspect of this research. 

3. Please also provide an updated Competing Interests Statement declaring this commercial affiliation along with any other relevant declarations relating to employment, consultancy, patents, products in development, or marketed products, etc. 

- We have included the following Competing Interest Statement to declare the commercial affiliations of GH and LD.

- KA declares that no competing interests exist.

GH has read the journal's policy and has the following competing interests:

* is an employee, board member, and shareholder of the company manufacturing the equipment used in this study

* is a named inventor of patent applications assigned to the company manufacturing the equipment used in this study.

LD has read the journal's policy and has the following competing interests:

* is an employee and shareholder of the company manufacturing the equipment used in this study

* is a named inventor of patent applications assigned to the company manufacturing the equipment used in this study. 

This does not alter our adherence to PLOS ONE policies on sharing data and materials.

4. Please include a copy of Table 2 which you refer to in your text on page 4.

- We apologize for the confusion on this matter. The article has only one table. In the conversion from Word to PDF, the second reference to Table 1 was changed to “Table 2.” We will ensure the conversion is correct. 

5. We note you have included a table to which you do not refer in the text of your manuscript. Please ensure that you refer to Table 1 in your text; if accepted, production will need this reference to link the reader to the Table.

- Please see the response to question 4. The manuscript only contains one table. 

REVIEWER #1:

1. What was the Duty cycle? Could this be explained in more detail.

- Thank you for this important question. We have replaced “Duty Cycle” with “% time active” in Table 1. Duty cycle is an engineering description of the relative period that a signal is present within a given period of time. A duty cycle of 100% indicates that a signal is present throughout a time period while a duty cycle of 50% indicates that the signal is present half of the time. We believe that the new descriptor, “% time active” is more clear. Thank you for pointing out the use of jargon. 

2. What statistical tests did you used?

- Thank you for this question. Our manuscript is largely descriptive and we did not perform statistical analysis. 

3. How does the active noise canceling device work? More details would give the reader a clearer understanding about the device and how it works.

- We are appreciative of the opportunity to provide further clarity on how this device works. We have added the following text near line 82.

“Utilizing the phenomenon of incident waves summing, active noise control was accomplished by generating sound waves within the incubator that are out of phase with a model of sound waves detected by the outside noise sensor as they are when they have passed through the walls of the incubator. The residual noise remaining when the environmental and the cancelling sound wave meet was detected by the residual noise sensor, which is used to continually refine the cancelling sound wave produced to maximize the cancellation.” 

4. Is this device approved by health authorities like the FDA?

- This device is subject to regulatory clearance by the FDA before clinical use may be permitted. However, as this study is preclinical (simulation), FDA clearance was not required.

5. If you use CPAP as white background noise, would you also register the same level of dB for alarms?

- We appreciate the chance to clarify this matter regarding combining a primary noise with a background noise. Measurements of the equivalent continuous sound level (Leq) of a primary noise and a background noise consider the contribution of all sounds during the measurement period. Background noise will increase the Leq over the measured time period.

Note that a background noise can affect the “detectability” of a primary tone. However, this is not reflected in the equivalent sound level, Leq. This would affect the Tone-to-noise-ratio and the Prominence ratio. However, these are not the subject of this study.

---

## [Decision Letter · Decision Letter 1]

12 Jun 2020

Incubator-based Sound Attenuation: Active Noise Control In A Simulated Clinical Environment

PONE-D-20-05746R1

Dear Dr. George Hutchinson,

We’re pleased to inform you that your manuscript has been judged scientifically suitable for publication and will be formally accepted for publication once it meets all outstanding technical requirements.

Kind regards,

Anne Lee Solevåg, M.D., Ph.D.

Academic Editor

PLOS ONE

Additional Editor Comments (optional):

Reviewers' comments:

Reviewer's Responses to Questions

**Comments to the Author**

1. If the authors have adequately addressed your comments raised in a previous round of review and you feel that this manuscript is now acceptable for publication, you may indicate that here to bypass the “Comments to the Author” section, enter your conflict of interest statement in the “Confidential to Editor” section, and submit your "Accept" recommendation.

Reviewer #1: All comments have been addressed

2. Is the manuscript technically sound, and do the data support the conclusions?

Reviewer #1: Yes

3. Has the statistical analysis been performed appropriately and rigorously? 

Reviewer #1: Yes

4. Have the authors made all data underlying the findings in their manuscript fully available?

Reviewer #1: Yes

5. Is the manuscript presented in an intelligible fashion and written in standard English?

Reviewer #1: Yes

6. Review Comments to the Author

Reviewer #1: (No Response)

7. PLOS authors have the option to publish the peer review history of their article (what does this mean?). If published, this will include your full peer review and any attached files.

Reviewer #1: Yes: Georg Schmolzer

---

## [Editor Report · Acceptance letter]

22 Jun 2020

PONE-D-20-05746R1 

Incubator-based Sound Attenuation: Active Noise Control In A Simulated Clinical Environment 

Dear Dr. Hutchinson:

I'm pleased to inform you that your manuscript has been deemed suitable for publication in PLOS ONE. Congratulations! Your manuscript is now with our production department. 

Kind regards, 

on behalf of

Dr. Anne Lee Solevåg 

Academic Editor

PLOS ONE